# An Ensemble Learning for Automatic Stroke Lesion Segmentation Using Compressive Sensing and Multi-Resolution U-Net

**DOI:** 10.3390/biomimetics10080509

**Published:** 2025-08-04

**Authors:** Mohammad Emami, Mohammad Ali Tinati, Javad Musevi Niya, Sebelan Danishvar

**Affiliations:** 1Department, Faculty of Electrical and Computer Engineering, University of Tabriz, Tabriz 51666-16471, Iran; 2College of Engineering, Design and Physical Sciences, Brunel University London, Uxbridge UB8 3PH, UK

**Keywords:** CT images, compressive sensing, segmentation, ensemble learning

## Abstract

A stroke is a critical medical condition and one of the leading causes of death among humans. Segmentation of the lesions of the brain in which the blood flow is impeded because of blood coagulation plays a vital role in drug prescription and medical diagnosis. Computed tomography (CT) scans play a crucial role in detecting abnormal tissue. There are several methods for segmenting medical images that utilize the main images without considering the patient’s privacy information. In this paper, a deep network is proposed that utilizes compressive sensing and ensemble learning to protect patient privacy and segment the dataset efficiently. The compressed version of the input CT images from the ISLES challenge 2018 dataset is applied to the ensemble part of the proposed network, which consists of two multi-resolution modified U-shaped networks. The evaluation metrics of accuracy, specificity, and dice coefficient are 92.43%, 91.3%, and 91.83%, respectively. The comparison to the state-of-the-art methods confirms the efficiency of the proposed compressive sensing-based ensemble net (CS-Ensemble Net). The compressive sensing part provides information privacy, and the parallel ensemble learning produces better results.

## 1. Introduction

A stroke is among the foremost causes of mortality and disability, impacting 15 million individuals worldwide each year [1]. Following a stroke, fifty percent of survivors have long-term disabilities [2]. Approximately 90% of strokes are ischemic, indicating that an obstructed artery results in tissue ischemia, infarction, and diminished cerebral blood flow [3]. A rupture or tear in a weakened blood vessel is the cause of a type II stroke. Research conducted by the Food and Drug Administration revealed that 13% of strokes in the United States are hemorrhagic, whereas 87% are ischemic. Age, gender, color, and ethnicity are all critical factors in determining the underlying causes of stroke [4].

The diagnosis of a stroke relies on comprehensive medical records, physical and neurological examinations, and brain imaging tests such as CT scans or magnetic resonance imaging (MRI) to exclude alternative diagnoses, including brain tumors and drug effects [5,6].

Conventional techniques for the automated identification and segmentation of ischemic stroke lesions include the use of a neuroradiologist. The traditional approach is the neuroradiologist visually detecting the infarct or shadow region and notifying the specialized physician of the nature of the occluded arteries [7]. Lesions, however, are heterogeneous, differing in morphology, dimensions, and anatomical position. Consequently, the interpretation and diagnosis of brain imaging are complex processes that need specialist expertise [8]. Moreover, the conventional analysis of stroke lesions by neuroradiologists is laborious, susceptible to errors, and imprecise [9]. Also, an erroneous diagnosis might have permanent repercussions. Moreover, the contrast of the acquired pictures may be insufficient, hindering the neuroradiologist’s ability to accurately diagnose the lesion and vascular structures owing to image noise [10]. Numerous research studies have lately emerged in the domain of automated ischemic stroke diagnosis, which we shall examine in detail below.

Sultanpour et al. [11] used four two-dimensional U-Net networks for the automated segmentation of ischemic stroke lesions. Outliers were eliminated during the pre-processing phase of the researchers’ model, and the dataset was enhanced by rotation, scaling, and the inclusion of noise. Subsequently, parallel U-Nets were trained using computed tomography perfusion (CTP) maps. The segmentation performance of the researchers’ model was evaluated using the Dice Similarity Coefficient (DSC) and volumetric similarity metrics, with the ISLES 2018 challenge dataset used for network training and assessment. Liu et al. [12] used CTP images for the automated segmentation of ischemic stroke lesions. The researchers employed generative adversarial networks (GANs) in their proposed approach to create diffusion-weighted imaging (DWI). The convolutional networks used by these researchers generally include a generator, a separator, and a divider. The researchers’ DSC is reported as 49% based on four-fold cross-validation. Doles et al. [13] used modified U-Net networks for the classification of ischemic stroke lesions. The researchers’ architecture was notable in three aspects. Initially, instead of amalgamating picture modalities into a single input, each modality is processed independently to use the distinct information more effectively. The network suggested by the researchers was intricately interconnected. Third, drawing inspiration from the Inception network model, these researchers enhanced the conventional U-Net modules by integrating two convolutional blocks in the input layers to optimize performance across different scales. The results of this research indicated that multimodal information surpassed simpler tactics. Wong et al. [14] introduced a novel framework using DWI from CTP maps to enhance the quality and precision of brain lesion segmentation. The researchers proposed a model comprising three components, based on end-to-end trained convolutional neural networks (CNNs). These researchers secured first place in the ISLES 2018 Image Challenge. Albert et al. [15] examined the automated segmentation of ischemic strokes using CTP and CT images. The researchers used the ISLES challenge dataset to achieve this objective. They used an enhanced U-Net architecture for picture segmentation. The assessment dataset included 94 lesion image instances for network training and 62 examples for network testing. The 10-fold cross-validation technique is used to assess the suggested network, yielding a final Dice Similarity Coefficient of 49% for the image segmentation presented by the researchers. A downside of the study is its inadequate performance in lesion detection. Ghanmat et al. [16] introduced a novel model utilizing deep learning techniques for segmenting ischemic stroke lesions. In this study, the researchers used Euclidean distance to compute the average distance of each pixel in the rightward and downward directions. They enhanced the DSC by 2.5% relative to previous research. Raju et al. [17] used U-Net networks for the automated segmentation and diagnosis of ischemic strokes. They introduced a U-Net-derived deep model that can concurrently and autonomously analyze all CTP maps. The DSC reported by these studies was roughly 42%. Sultanpour et al. [18] introduced a deep neural network for the automated segmentation of ischemic stroke lesions. This research used a modified design of the U-Net network. Multi-scale CNNs and CNN-based shortcut links were used inside this network. Their concluding study indicated a detection sensitivity coefficient (DSC) of 68%.

According to previous research reviews, each of these studies has its advantages and disadvantages; however, the main issue with the automatic segmentation of DSC stroke lesions is the low accuracy rate, which is less than 68%. Also, the previous works lack a procedure for concealing private information. Moreover, many of them consider a single learner to train and learn the parameters. The proposed network in this study aims to address these challenges by employing a combination strategy that utilizes ensemble learning and compressive sensing. Moreover, it provides a multi-resolution modified U-shape network to obtain a systematic network architecture for automatic segmentation of brain stroke lesions.

The contributions to this study can be summarized as follows:It provides a network to protect the private content of data by utilizing the compressed version of the input medical image.The proposed network provides an ensemble approach to train the compressed version of the input image.It represents a combination of compressive sensing and an ensemble of parallel learners to extract the stroke lesion.It provides a novel ensemble multi-resolution U-shaped network for segmenting the medical stroke CT dataset. The term multi-resolution, as used in this study, describes the convolutional structure of the model in which multiple branches with distinct kernel sizes operate in parallel. This allows the network to extract both fine-grained and coarse-grained spatial features from CT scans, which is particularly beneficial for capturing stroke lesions with varying sizes and textures.The proposed network utilizes a channel of perfusion maps, including CBV, CBF, MTT, Tmax, and CT slice, to efficiently extract the stroke lesion.

The subsequent sections of this work are structured as follows: Section 2 examines the used database, deep convolutional networks, and the U-Net architecture. Section 3 delineates the recommended methodology of the current investigation. Section 4 outlines the simulation outcomes using the suggested method. Section 5 pertains to the conclusion.

## 2. Materials and Methods

This section provides a comprehensive examination of the database used, including details of data capture and collection. Subsequently, deep convolutional networks will be elucidated in conjunction with the U-Net architecture.

### 2.1. ISLES Database

The stroke lesion CT database includes the CT and perfusion maps of CBV, CBF, MTT, and TTP. Each map consists of 156 images in the .nii format. The pre-processing stage begins with converting the image format to JPEG. The 94 CT cases of patients included different numbers of slices in the CT map, as well as maps of CBV, CBF, MTT, and TTP. There are 502 images of each map. There are 502 mask images of the CT maps created by the doctors. The corresponding CT maps and CT perfusion maps of the correct masks are used to obtain the network weights [19].

Details of the database used are shown in Table 1. The samples of CT slices, corresponding masks, and the borders of the stroke lesion on CT are shown in Figure 1.

### 2.2. CS Theory

This section introduces the theoretical foundation of compressive sensing (CS) and its application in signal representation and reconstruction [20].

In principle, CS employs a quasi-optimal measurement technique that gathers all raw signal data and generates an efficient, dimensionally reduced signal representation. Compressed sensing theory enables the reconstruction of a sparse signal using far fewer samples than those required by the Shannon–Nyquist sampling theorem [20]. Discrete wavelet transformation (DWT) and discrete cosine transformation (DCT) are two signal transformations that effectively convert numerous nonsparse signals into sparse signals. Consequently, nonsparse signals may also be analyzed using the CS theory, provided that the sparsity of the transformed signal is guaranteed. The compressed output, *Y ∈ R^M^*, is produced by using *X ∈ R^N^* as the input signal and *Φ ∈ R^M^^×N^* as the measurement matrix.(1)Y=ΦX

When the measurement matrix fulfils the restricted isometry property (RIP), the result is M ≪ N, signifying that the output observation vector has much lower dimensionality than the input signal. The RIP stipulates that the measurement matrix for any strictly sparse vector must satisfy the following criteria:(2)$$1-\delta \leq \frac{{\left\|\Phi X\right\|}_{2}^{2}}{{\left\|X\right\|}_{2}^{2}}\leq 1+\delta$$
where the RIP constant, represented by δ, varies from 0 to 1. The extent of signal compression is represented by the compression ratio (CR), defined as follows:(3)$$CR=\frac{N-M}{N}\times 100\%$$

Thus, a large CR will result from *M* being much less than *N*. Altering the dimensions of the measurement matrix may provide variable levels of compressed acquisitions, provided the RIP condition is met. Finally, specific techniques, such as orthogonal matching pursuit (OMP) [20] and L1 norm minimization [21], may achieve an accurate reconstruction of *X* [21].

### 2.3. Convolutional Networks

Convolutional neural networks are among the most significant deep learning architectures. This network was created by Mr. Hubble and Weissl in 1990, drawing inspiration from the human visual cortex. The primary application of these networks was the recognition of handwritten digits, yielding promising results. These networks were computationally impractical during that period due to insufficient graphics hardware and were subsequently discontinued. In recent years, the proliferation and enhancement of graphics processing units have revitalized the popularity of deep convolutional networks, which are now employed across diverse application domains. The resurgence of these networks can be assessed post-2012. The primary applications of convolutional neural networks are in object recognition, speech processing, and facial recognition. Convolutional networks, similar to neural networks, consist of layers of neurons equipped with weights and biases that can be optimized through learning. These networks consist of various blocks, as illustrated in Figure 2. The blue block in the illustration represents the convolution layer, which features a nonlinear function. The red block signifies a pooling layer, while the green block denotes a fully connected layer. The convolutional layer is the fundamental component of the CNN. This layer, composed of various filters, can rotate on the input signal to generate the output feature map. The quantity of these filters is expressed as a power of 2, ranging from 32 to 4096. Increasing the number of filters enhances the network’s capacity; however, this can lead to overfitting. The dimensions of the filters are specified as 3 × 3, 5 × 5, and 7 × 7, presented in a square format. The number of learnable parameters can be diminished by selecting smaller filters [22].

The step constitutes an additional parameter within the convolutional layer. This parameter determines the persistence of filtering in network computations. Typically, in convolutional networks, the dimensionality of this parameter is considered to be 2. Likewise, the association of the convolutional layer can be articulated as follows:(4)$$y_{k}=\sum_{n=0}^{N-1}x_{n}h_{k-n}$$

In the aforementioned relationship, the variables *x*, *h*, and *N* denote the input, the filter, and the number of elements in the output vector y, respectively. Activation functions are applied after the convolutional layer within the network. These functions induce nonlinearity within the network. Among the prevalent activation functions in convolutional networks are ReLU and Leaky ReLU. The ReLU function nullifies negative values in the network and transmits values exceeding zero to the output. This function is utilized by default in convolutional networks because of its simplicity and the enhancement of algorithm speed and network convergence [22]. The merging layer is an additional component in convolutional networks utilized to diminish the spatial dimensions of feature maps. This block lacks a training parameter and functions similarly to sampling, acting as a filter within the convolutional layer. The most prevalent filters are the max-merge and mean-merge layers. A preconfigured window measuring (3 × 3) is utilized for max-merge, which is traversed across the image to identify the maximum value, while the remaining values are set to zero. This layer reduces the feature dimensions. The ultimate layer of the convolutional network is the fully connected layer, wherein the classification of the selected or extracted features occurs. This layer is utilized in all classical neural networks. This layer transforms the chosen feature matrix into a vector and categorizes it into various designated classes. This layer computes the probability distribution of the output classes, which is articulated as follows:(5)$$p_{i}=\frac{e^{x_{j}}}{\sum_{1}^{k}e^{x_{K}}}\;\mathrm{for}\;j=1,\;\ldots k$$

In the aforementioned relationship, the variables *x* and *p* denote the network’s input and output value, respectively, which range between 0 and 1, with their sum equating to 1.

The aforementioned layers are recognized as the primary layers in convolutional networks. Indeed, additional layers are employed in these networks, which will be explained in more detail subsequently. Additional frequently utilized layers comprise the random deletion layer and the batch normalization layer. The dropout layer is employed to disregard the performance of specific neurons within the network. This layer inhibits the overfitting phenomenon in the network. The batch normalization layer is employed to standardize the data within the network. This layer diminishes internal covariance, consequently enhancing the training speed of the network. The subsequent relationship pertains to the efficacy of the batch normalization layer:(6)$$\begin{array}{l} \mu_{B}=\frac{1}{n}\sum_{i=1}^{n}y_{i}^{\left(l-1\right)}\\ \sigma_{B}^{2}=\frac{1}{n}\sum_{i=1}^{n}{\left(y_{i}^{\left(l-1\right)}-\mu_{B}\right)}^{2}\\ {\overset{\wedge }{y}}^{\left(l-1\right)}=\frac{y^{\left(l-1\right)}-\mu_{B}}{\sqrt{\left(\sigma_{B}^{2}+\varepsilon \right)}}\\ z^{\left(l\right)}=\gamma^{\left(l\right)}{\overset{\wedge }{y}}^{\left(l-1\right)}+\beta^{\left(l\right)} \end{array}$$

In the aforementioned equation, $\mu_{B}$ represents the mean, while $\sigma_{B}^{2}$ denotes the variance of the class. Furthermore, this equation indicates that ε represents a small numerical constant, l the number of each layer, $y^{\left(l-1\right)}$ the input vector for the normalizer layer, $z^{\left(l\right)}$ the output vector for a neuron, $\gamma^{\left(l\right)}$ the corresponding scale, and $\beta^{\left(l\right)}$ the learning rate adjustment in the network [23].

### 2.4. U-Net

Due to the significance of pictures and videos as information sources for several applications and systems, image processing and pattern recognition are regarded as critical domains within computer science. The U-Net architecture is a notable design in this domain. This architecture is a convolutional neural network intended for image segmentation. This architecture has revolutionized medical image processing due to its versatility and enhanced precision in segmentation, even with limited data. A distinguishing feature of U-Net is the integration of Skip Connection layers, which allow the network to retain local and detailed information at each stage of image processing. This capacity will enable U-Net to recognize complex images with high accuracy and efficiency, producing accurate results. The U-Net architecture has a symmetrical design that minimizes the number of network parameters and mitigates overfitting. This capability enables U-Net to provide superior outcomes with less data and time requirements [24].

U-Net is designed for picture segmentation, with a U-shaped architecture that comprises a downsampling pathway and an upsampling pathway. This architecture enables the convolutional network to acquire and integrate information at varying degrees of granularity. This is crucial for precisely segmenting minor regions and intricate features in photos.

The downsampling route of U-Net emphasizes the extraction of significant characteristics from the picture. In contrast, the upsampling route concentrates on rebuilding the image from these features to maintain essential details. The downsampling component comprises multiple layers that utilize convolutional and pooling processes. These layers progressively extract visual characteristics and diminish the image size. At each level, the quantity of feature channels is augmented to gather more information about the picture.

The incremental segment is the inverse of the preceding structure, using techniques such as up-convolution or deconvolution and concatenation to progressively augment the picture dimensions while integrating the features acquired during the reduction phase with the incremental layers. This enables the network to maintain spatial information and enhance segmentation accuracy. Ultimately, a convolutional layer restores the feature map to the original picture dimensions, categorizing the image pixels into distinct classes.

The down-sampling pathway adheres to the conventional structure of convolutional networks. The process involves successive 3 × 3 convolutions (unpadded), each followed by a modified ReLU and a 2 × 2 Max Pooling operation with a stride of 2 to reduce the sample size. During each down-sampling phase, the number of picture channels is increased twofold. This segment of the network facilitates the extraction of more intricate characteristics while maintaining reduced spatial information. The upsampling path layers are the inverse of the downsampling route. This segment of the network entails deconvolution, which enlarges the picture size by a factor of two. Following each deconvolution process, a concatenation occurs with the matching mapped features from the downsampling pathway to restore spatial information. This procedure facilitates the retrieval of information lost during downsampling, enabling the network to achieve enhanced spatial precision in the final segmentation. After the architecture, a 1 × 1 convolution layer transforms the feature map into the requisite number of classes for segmentation. This layer is used to ascertain the classification of each pixel in the final segmented picture. The unit architecture is shown in Figure 3 [24].

## 3. Proposed Model

This section outlines the proposed methodology of this work, including details on data preparation, architecture, and training and testing procedures.

### 3.1. Pre-Processing of Data

The Hounsfield unit ranges of the pixel intensities are preprocessed with a bandpass window of [0, 150] to remove noise and unwanted ranges of this unit. The dimension of the images after the resize stage of pre-processing is 256 × 256. The augmentation is performed as the next step of the pre-processing stage. Horizontal flip, vertical flip, and rotation of (0,25) degrees are performed to increase the number of images and the corresponding mask images. Normalization is the third step in this stage, which limits the range of intensities to the [0, 1] interval.

### 3.2. Proposed Compressive Sensing-Based Ensemble Net (CS-Ensemble Net)

The schematic overview of the proposed method is illustrated in Figure 4. The CT images from different modalities are used as a database to detect stroke lesions automatically. As demonstrated, after the pre-processing stage, the obtained images are fed into the proposed CS-Ensemble Net for training. The training phase of the proposed network is conducted using images, maps, and their corresponding masks. The trained network can be used to extract the stroke lesion.

### 3.3. Proposed CS-Ensemble Net Architecture

The training is performed using an ensemble learning. The proposed network architecture combines compressive sensing with a 2 × 2-resolution U-Net and a 3 × 3-resolution Modified U-Net. The teaching of the network’s weights is the sum of the two cost functions of the two networks. The input images are imposed on the compressive-sensing module.

This module consists of a two-dimensional convolution, with the number of filters changed based on the compression ratio. The output of the CS module is applied to a two-resolution UNet. Another component of the Ensemble network is a three-resolution UNet, which receives image arrays from the CT stroke database. Figure 5 demonstrates the two networks of the proposed Ensemble network. The first one consists of a CS module and a two-resolution UNet. The second one is a three-resolution modified UNet. Figure 6 illustrates the details of the proposed network. Further information regarding the encoder and decoder blocks of the multi-res modified UNets can be seen in this figure. According to Figure 6, the 2-Res and 3-Res U-shape architectures of the deep network contain encoder and decoder blocks. The details of the layer types, the number of weights, and the output dimensions of the layers are explained in Table 2, Table 3 and Table 4.

Instead, it ensures spatial dimension alignment between the compressed output and the expected input size of the subsequent U-Net branch. The compression is preserved in the feature space, where only essential structural representations are passed forward, maintaining the privacy-aware benefits of the CS process.

Table 3 describes a conventional U-Net without any multi-resolution characteristics, making the label “2-resolution U-Net” potentially misleading. The details of the CS module are represented in Table 2. The number of filters varies depending on the CT ratio. As the number of filters in this specific two-dimensional convolution increases, the compressive sensing ratio becomes high. Table 3 shows the details of the first part of the ensemble network. It consists of several two-dimensional convolutional and transposed convolutional layers. Layers of max pooling and concatenation are needed to complete this U-shaped network. There are several layers used for constructing the other part of the ensemble network. The total number of trainable parameters for this 3-resolution UNet is more than 84 million parameters, as shown in Table 4. The total number of trainable parameters for the first part exceeds 31 million.

In this study, the term “resolution” refers specifically to the number of encoder–decoder depth levels (i.e., downsampling and upsampling stages) within the U-Net architecture. The 2-resolution U-Net contains two levels of encoding and decoding layers. Each level includes standard convolutional blocks using 3 × 3 kernel sizes, a stride of 1, and a dilation rate of 1, followed by 2 × 2 max pooling for downsampling and transpose convolution for upsampling. The 3-resolution U-Net consists of three such levels, similarly built with 3 × 3 kernels and no dilation. This deeper structure enables a broader, more comprehensive representation of context at multiple scales. All architectural parameters, including kernel size, stride, and activation functions used in both configurations, are detailed in Table 2, Table 3 and Table 4. This clarification aims to remove any ambiguity regarding the use of the term “resolution” in our proposed model.

The term “multi-resolution” in this work refers to the use of multiple encoder–decoder levels in the U-Net architecture, where each level captures different scales of spatial information. Each level includes two 2D convolutional layers with 3 × 3 kernels, followed by 2 × 2 max-pooling for downsampling. The decoder path uses 2 × 2 transpose convolution (upsampling) with concatenation of skip connections. This design enables the network to integrate low-level details and high-level context hierarchically. The full architectural specifications, including the number of filters and convolutional kernel parameters, are presented in Table 2, Table 3 and Table 4 for reproducibility and clarity.

The proposed CS-Ensemble Net incorporates two parallel branches: The first branch includes a compressive sensing module followed by a 2-resolution U-Net, the second is a 3-resolution modified U-Net operating independently. Both networks are trained jointly using a composite loss function, defined as a weighted average of the Dice losses from each branch. During inference, the pixel-wise softmax probability maps from both U-Nets are averaged to produce the final segmentation mask. This strategy leverages the strengths of both shallow and deep pathways, yielding more accurate and stable segmentation results.

### 3.4. Training and Evaluation of the CS-Ensemble Net

During the training procedure, several parameters, including the optimizer, learning rate, number of layers, and loss function, should be selected and tuned. The optimal parameters are reported in Table 5. The training is performed using a 10-fold cross-validation procedure.

## 4. Experimental Results

This section presents the outcomes of the suggested methodology. The Club platform was utilized for all simulations in this study, and its features include a Tesla K90 GPU with 25 GB of RAM.

The segmentation results were evaluated based on the DSC [25] and Intersection over Union (IoU) [26]. The correlations among the considered criteria are as follows:(7)$$DSC=\frac{|A⋂B|\times 2}{\left|A\right|+|B|}$$(8)$$IoU=\frac{\left|A\cap B\right|}{\left|A\cup B\right|}$$

According to the above formulas, A: The area (Segment) predicted by the model or method under consideration, B: The reference area or Ground Truth, ∣A∩B∣: The number of common pixels between A and B, ∣A∣ and ∣B∣: The total number of pixels in each of the areas A and B is displayed separately. Also, ∣*A*∪*B*∣: The total number of pixels combined from A and B (i.e., the total area covered by A or B or both). In addition, various criteria, including accuracy, precision, and sensitivity, have been used in this research to evaluate the suggested model, whose calculation formula is presented in the following relationships.(9)$$Accuracy=\frac{T_{p}+T_{N}}{T_{p}+F_{P}+T_{N}+F_{N}}$$(10)$$Precision=\frac{T_{p}}{T_{p}+F_{P}}$$(11)$$Sensitivity=\frac{T_{p}}{T_{p}+F_{N}}$$

As per the previously mentioned correlations, samples in categorization classes are represented by *T_P_*, *T_N_*, *F_P_*, and *F_N_*, which stand for true positive, true negative, false positive, and false negative, respectively.

In this section, the simulation results of the proposed Cheb-MA are presented. Our framework is implemented on a laptop (Manufacturer: Dell, Austin, TX, USA) with 16 GB of RAM, a 2.8 GHz Core i7 CPU, and a GeForce GTX 1050 GPU. The proposed network is trained using the Google Colaboratory Pro platform.

The loss function regarding the dice loss is represented in Figure 7. As can be seen, 400 iterations are required for the convergence of the CS-Ensemble 3-res UNet. The plot compares the fluctuations of three different methods. Figure 8 illustrates the accuracy of our proposed CS-Ensemble Net, the Ensemble Modified-Res UNet, and the Multi-Res UNet. This figure, along with the fluctuations in dice loss, confirms the efficiency of our proposed method. The segmentation results, as evaluated by the accuracy, sensitivity, dice coefficient, and mean IoU metrics, corresponding to the different perfusion maps, are reported in Table 6. The segmentation results, considering all perfusion maps and corresponding CT slices, are shown in this figure. Table 7 illustrates the segmentation results according to different compressive sensing ratios.

The training procedure was performed considering all the perfusion maps as well as the CT slices. Figure 9 illustrates the obtained Dice coefficients for various compressive sensing ratios. Moreover, the computational complexity is presented in terms of time per epoch in seconds, compared to the considered compressive sensing ratios. According to this figure, an acceptable computational burden with good performance in terms of dice coefficient has been acquired with a CS ratio of 80%. Figure 10 illustrates the segmented border, colored green, in comparison to the mask border, colored cyan, according to the 80% confidence score (CS) ratio. Figure 11 illustrates the segmentation borders for a 100% CS ratio, marked in purple, in comparison to the masks shown in cyan. Figure 12 compares the Dice Coefficient for different resolutions of the encoder and decoder in the CS-Ensemble Net. Additionally, it illustrates the computational complexity of the incremental trend in resolution, measured in terms of time per epoch in seconds. According to this figure, the best resolution with the least computational burden has been achieved in three resolution types of encoder and decoder.

Figure 13 illustrates the noisy perfusion map, and Table 8 explains the segmentation results, along with performance metrics. The results demonstrate the network’s efficiency in conditions of noisy images. Table 9 presents the evaluation metrics for comparing the results with state-of-the-art methods. It shows the efficient performance of the proposed CS-Ensemble Net. The term “Ensemble Net” used in these tables refers to the dual-branch architecture consisting of both 2-resolution and 3-resolution U-Nets, but without the CS module. That is, the input images are passed directly to both branches without any preprocessing for compression. In contrast, the “CS-Ensemble Net” includes the CS-based compressed input path for one of the branches, as described in Section 3.3.

The proposed CS-Ensemble Net includes over 115 million trainable parameters due to its dual-branch structure. Training each fold (in 10-fold cross-validation) required approximately 3.5 h, and inference time per CT volume was about 1.2 s. While our goal was maximizing segmentation accuracy, future work will explore model compression for real-time clinical feasibility. The proposed CS-ensemble requires a large number of parameters to be trained. It is recommended that future work consider addressing this challenge. Besides, the proposed network in this study should be tried with other medical datasets to report the evaluation metrics. Another critical point is to test the compressive sensing part in conjunction with the previously proposed deep network to compare the results of the network when presented with compressed input.

## 5. Discussion

Automatic detection and segmentation of ischemic stroke lesions play a crucial role in early diagnosis and timely treatment decisions, which are vital for reducing mortality and long-term disability in patients. Manual interpretation of CT perfusion maps by radiologists is labor-intensive and subject to inter-observer variability. In contrast, automated tools offer the potential for reproducible and time-efficient analysis in clinical workflows. The proposed CS-Ensemble Net addresses several limitations of existing methods by integrating compressive sensing and multi-resolution ensemble U-Net architectures. This approach enables the network to operate on compressed data while preserving key anatomical structures, thereby enhancing patient data privacy without compromising accuracy. Compared to prior methods such as R2U-Net, SLNet, and DenseUNet, the CS-Ensemble Net demonstrates improved Dice coefficients and IoU scores across multiple CT perfusion modalities. The ensemble design of dual-resolution U-Nets ensures a balance between global context extraction and local detail preservation, which is essential for detecting lesions with varied shapes and scales. Another notable strength of the model is its resilience to noise. As shown in Table 8 and Figure 13, the CS-Ensemble Net maintains high segmentation performance even under noisy conditions, simulating the presence of imaging artifacts or low signal-to-noise ratios commonly encountered in real-world scenarios. This robustness indicates its reliability in clinical deployment. Overall, the proposed model achieves a practical compromise between segmentation accuracy, computational efficiency, and data privacy. It offers promising utility for integration into computer-aided diagnosis systems for assessing acute ischemic stroke.

The proposed multi-resolution CNN model not only excels at classifying strokes from brain CT images, but it also has significant potential for real-world applications. Because it has a lightweight structure and can make quick inferences, it can be easily integrated into radiology workflows, especially in emergency departments where rapid diagnosis is crucial. The model can also help radiologists make better decisions, especially in resource-constrained settings, by facilitating the interpretation of images and manually facilitating the easier adoption of PACSs (Picture Archiving and Communication Systems) by hospitals. In the future, we will focus on prospective validation using larger, real-time clinical datasets and creating user-friendly interfaces to make it easier for hospitals to use PACS (Picture Archiving and Communication Systems).

In recent advancements in the field of deep learning and machine learning, numerous studies have focused on innovative approaches to enhance diagnostic methods in various medical fields, including the detection of brain stroke. For instance, several works have demonstrated the power of computational models, such as level-set methods, to estimate tissue material parameters and improve clinical imaging analysis [30]. In line with this, federated learning techniques have been explored for heterogeneous environments to speed up machine learning processes, particularly in healthcare data analysis [31]. Moreover, artificial intelligence has been shown to augment decision-making processes, especially in fields like education, to address diversity and inclusivity issues [32]. The role of AI and machine learning in optimizing supply chains and enhancing financial market analysis has also been extensively studied [33,34]. Furthermore, machine learning has become an indispensable tool in areas like pedestrian behavior analysis and structural risk assessments [35,36], as well as in optimizing energy scheduling within smart grids [37,38,39,40]. The integration of deep learning and machine learning with generative AI has been a significant development, enabling improved model training and adaptive learning strategies [39]. Additionally, deep learning algorithms, particularly convolutional neural networks (CNNs) and recurrent neural networks (RNNs), are being increasingly employed for real-time anomaly detection and risk assessment in various sectors [41]. These studies emphasize the remarkable capability of AI and machine learning in transforming medical diagnostics, including brain stroke detection, by leveraging advanced data-driven methodologies. Notably, the application of structural equation models, multi-agent learning strategies, and reinforcement learning has led to more accurate predictions and efficient decision-making processes across diverse domains [42,43,44,45]. The continued evolution of AI, from generative AI to more innovative models, is paving the way for new advancements in stroke detection and overall healthcare solutions [46,47,48]. Additionally, AI’s contribution to various engineering fields, such as in the automotive and pharmaceutical industries, further demonstrates the extensive applicability of machine learning and deep learning models [49,50,51].

## 6. Conclusions

In this paper, a novel compressive sensing-based ensemble network is proposed. It paves the way to protect patient information related to abnormalities. The ensemble of two parallel learners presents a network with notable improvements in various segmentation performance metrics. The proposed network comprises a compressive sensing part and two modified U-shaped networks with multi-resolution encoders and decoders. The ablation studies with different combinations of perfusion maps from the ISLES 2018 dataset yield efficient results regarding the compressive sensing of 80% and 3-resolution type of encoder and decoder. The best performance is achieved using a channel of CBV, CBF, MTT, Tmax, and CT slice as the input image, compared to other state-of-the-art networks.

## Figures and Tables

**Figure 1 biomimetics-10-00509-f001:**
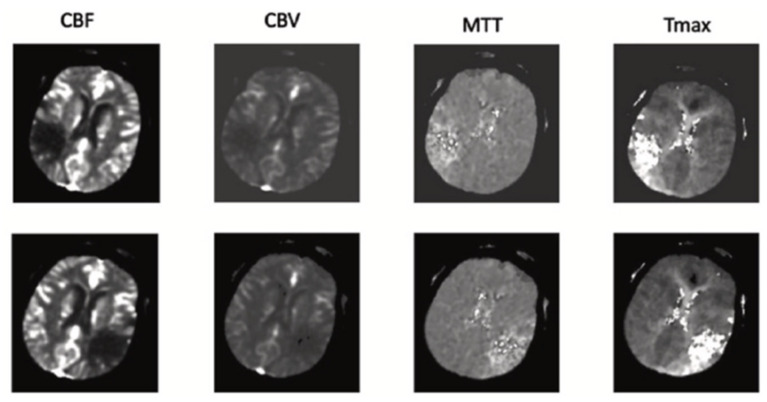
CTP maps to the ISLES database’s corresponding image [19].

**Figure 2 biomimetics-10-00509-f002:**
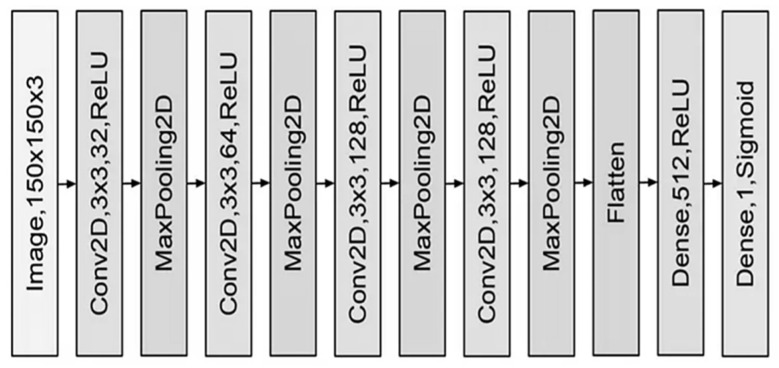
An elementary illustration of a convolutional neural network.

**Figure 3 biomimetics-10-00509-f003:**
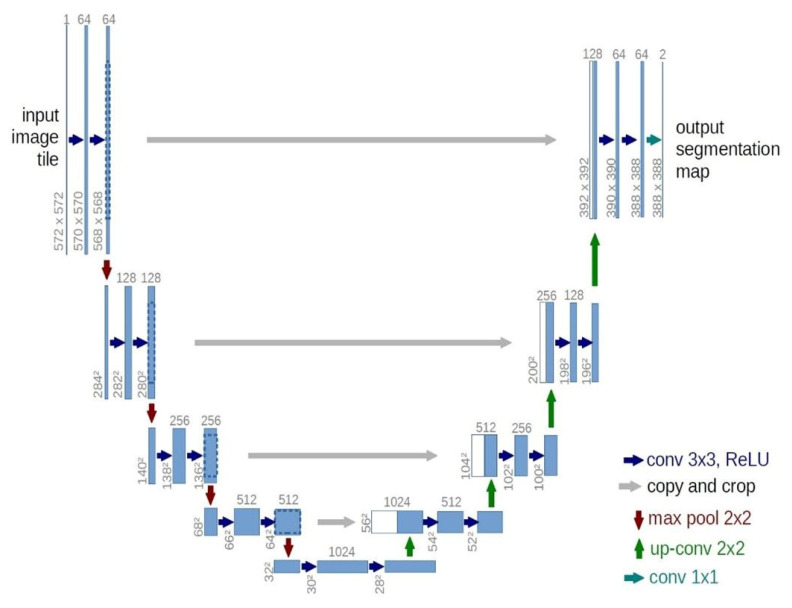
Architecture of a standard unit network.

**Figure 4 biomimetics-10-00509-f004:**
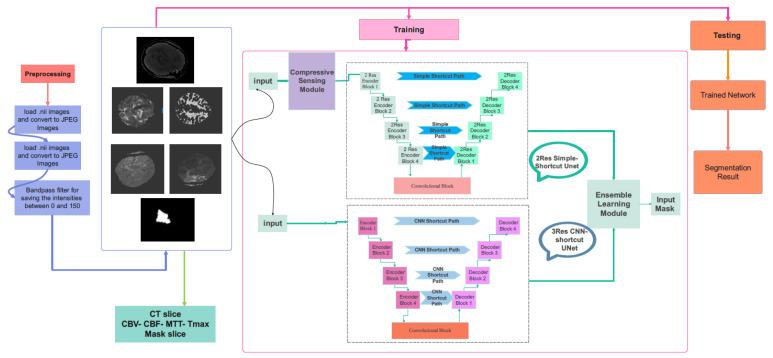
The overview of the proposed method (CS-Ensemble Net). The proposed model includes image preprocessing, model training and evaluation, and segmentation.

**Figure 5 biomimetics-10-00509-f005:**
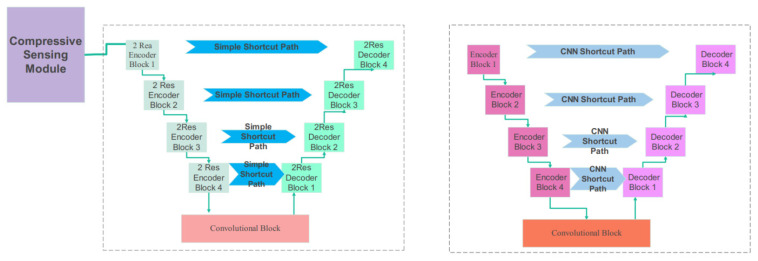
The proposed network architecture. The proposed model includes image preprocessing, model training and evaluation, and segmentation.

**Figure 6 biomimetics-10-00509-f006:**
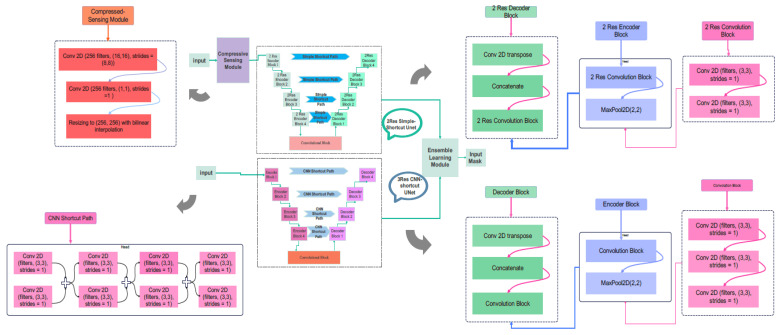
Overall architecture of the proposed multi-resolution convolutional neural network (CNN) model for stroke detection from brain CT images. The design consists of multiple parallel convolutional branches, each operating with different kernel sizes (3 × 3, 5 × 5, and 7 × 7), enabling the model to extract features at varying spatial scales.

**Figure 7 biomimetics-10-00509-f007:**
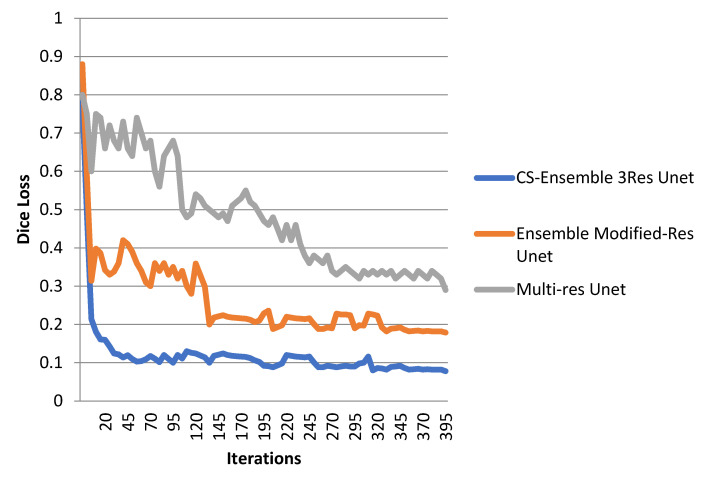
Dice Loss fluctuations for the three methods.

**Figure 8 biomimetics-10-00509-f008:**
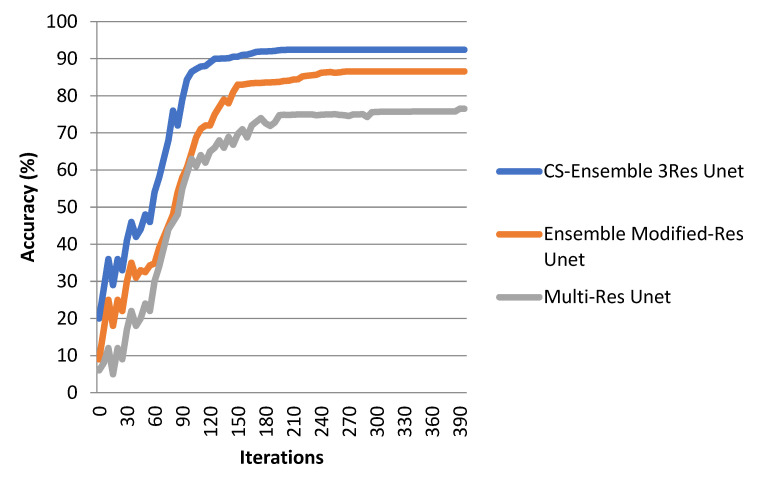
Accuracy fluctuations for three methods.

**Figure 9 biomimetics-10-00509-f009:**
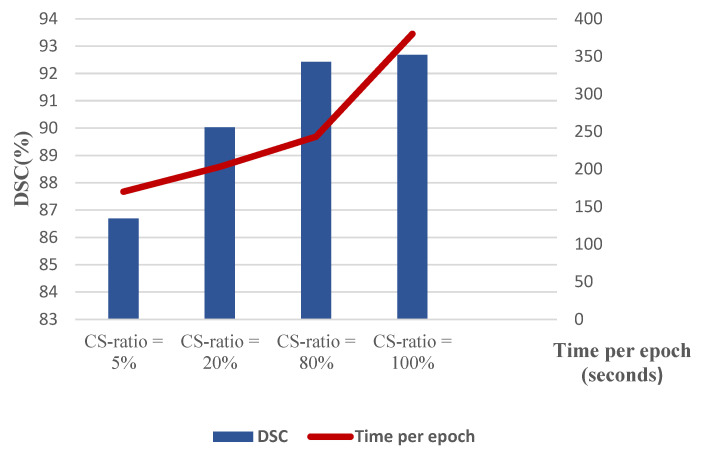
DSC% and computational complexity with 3-Res Conv 2-D.

**Figure 10 biomimetics-10-00509-f010:**
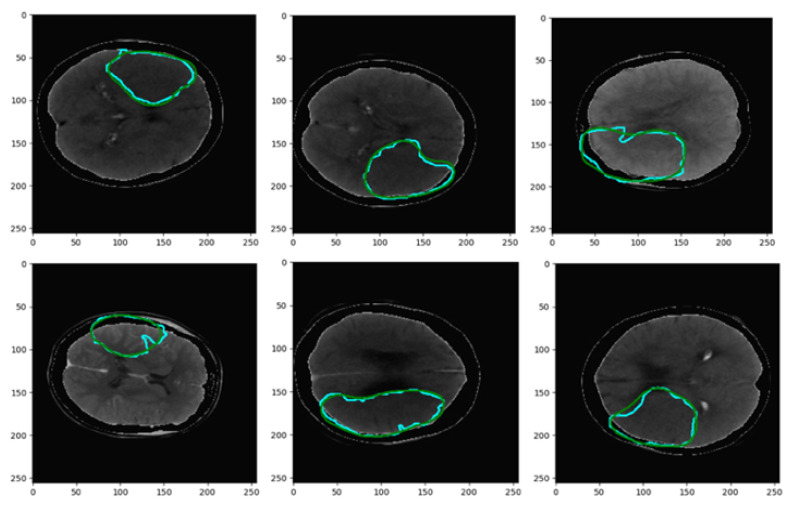
Segmentation results for a CS ratio of 80%. The cyan border indicates the mask, and the green indicates the result.

**Figure 11 biomimetics-10-00509-f011:**
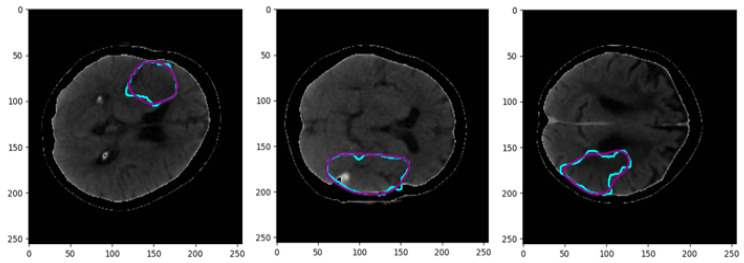
Segmentation results for a CS ratio of 100%. The cyan border represents the mask, and the purple border represents the result.

**Figure 12 biomimetics-10-00509-f012:**
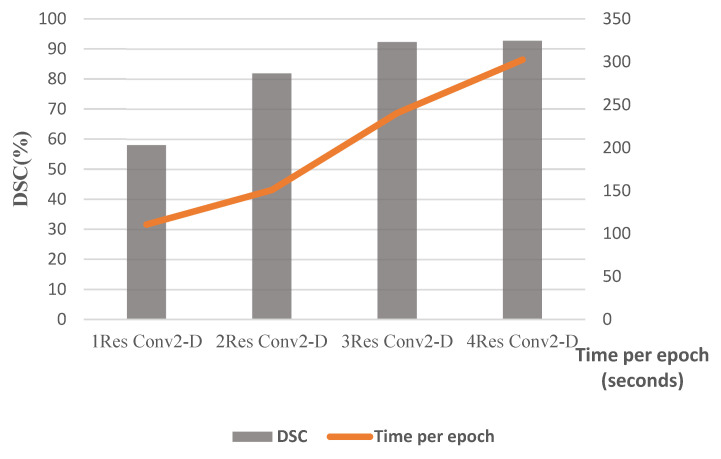
DSC% and computational complexity with CS ratio = 80%.

**Figure 13 biomimetics-10-00509-f013:**
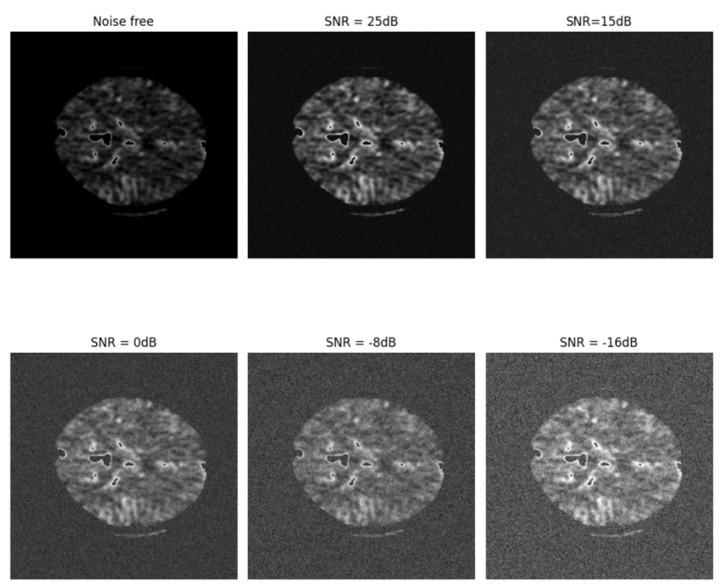
The noisy CT slice with varying signal-to-noise ratios (SNRs).

**Table 1 biomimetics-10-00509-t001:** The details of the stroke CT perfusion database.

CT Stroke Lesion	Category	Total Images	Dimension	Number of Train Images	Number of Test Images
**1**	CBV	502	256*256	450	52
**2**	CBF	502	256*256	450	52
**3**	MTT	502	256*256	450	52
**4**	Tmax	502	256*256	450	52
**5**	CT	502	256*256	450	52

**Table 2 biomimetics-10-00509-t002:** The selected optimal parameters for the design of the proposed network architecture.

Layer	Layer Name	Activation Function	Output Dimension	Size of Kernel	Strides	Number of Kernels	Number of Weights
1	Convolution 2-D	ReLU	(16, 31, 31, 256)	16 × 16	8 × 8	256	65,792
2	Convolution 2-D	ReLU	(16, 31, 31, 256)	1 × 1	1 × 1	256	65,792
3	Resizing	-	(16, 256, 256, 256)	-	-	-	0
	Total number of parameters						131,584

**Table 3 biomimetics-10-00509-t003:** The optimal parameters for the proposed network design were selected.

Layer	Layer Name	Activation Function	Output Dimension	Size of Kernel	Stride Shape	Number of Kernels	Number of Weights
1	Conv2-D	ReLU	(16, 256, 256, 64)	3 × 3	1 × 1	64	640
2	Conv2-D	ReLU	(16, 256, 256, 64)	3 × 3	1 × 1	64	36,928
3	MaxPooling 2-D	-	(16, 128, 128, 64)			64	0
4	Conv2-D	ReLU	(16, 128, 128, 128)	3 × 3	1 × 1	128	73,856
5	Conv2-D	ReLU	(16, 128, 128, 128)	3 × 3	1 × 1	128	147,584
6	MaxPooling 2-D	-	(16, 64, 64, 128)			128	0
7	Conv2-D	ReLU	(16, 64, 64, 256)	3 × 3	1 × 1	256	295,168
8	Conv2-D	ReLU	(16, 64, 64, 256)	3 × 3	1 × 1	256	590,080
9	MaxPooling 2-D	-	(16, 32, 32, 256)			256	0
10	Conv2-D	ReLU	(16, 32, 32, 512)	3 × 3	1 × 1	512	1,180,160
11	Conv2-D	ReLU	(16, 32, 32, 512)	3 × 3	1 × 1	512	2,359,808
12	MaxPooling 2-D	-	(16, 16, 16, 512)			512	0
13	Conv2-D	ReLU	(16, 16, 16, 1024)	3 × 3	1 × 1	1024	4,719,616
14	Conv2-D	ReLU	(16, 16, 16, 1024)	3 × 3	1 × 1	1024	9,438,208
15	Conv2-D transpose	ReLU	(16, 32, 32, 512)	2 × 2	2 × 2	512	2,097,664
16	concatenate		(16, 32, 32, 1024)			-	0
17	Conv2-D	ReLU	(16, 32, 32, 512)	3 × 3	1 × 1	512	4,719,104
18	Conv2-D	ReLU	(16, 32, 32, 512)	3 × 3	1 × 1	512	2,359,808
19	Conv2-D transpose	ReLU	(16, 64, 64, 256)	2 × 2	2 × 2	256	524,544
20	concatenate		(16, 64, 64, 512)			-	0
21	Conv 2-D	ReLU	(16,64,64,256)	3 × 3	1 × 1	256	1,179,904
22	Conv2-D	ReLU	(16, 64, 64, 256)	3 × 3	1 × 1	256	590,080
23	Conv2-D transpose	ReLU	(16, 128, 128, 128)	2 × 2	2 × 2	128	131,200
24	concatenate		(16, 128, 128, 256)			-	0
25	Conv 2-D	ReLU	(16, 128, 128, 128)	3 × 3	1 × 1	128	295,040
26	Conv2-D	ReLU	(16, 128, 128, 128)	3 × 3	1 × 1	128	147,584
27	Conv2-D transpose	ReLU	(16, 256, 256, 64)	2 × 2	2 × 2	64	32,832
28	concatenate		(16, 256, 256, 128)			-	0
29	Conv 2-D	ReLU	(16, 256, 256, 64)	3 × 3	1 × 1	64	73,792
30	Conv2-D	ReLU	(16, 256, 256, 64)	3 × 3	1 × 1	64	36,928
31	Conv2-D	ReLU	(16, 256, 256, 1)	2 × 2	2 × 2		65

**Table 4 biomimetics-10-00509-t004:** Full architecture of the 3-resolution U-Net is detailed below, including kernel sizes, strides, output dimensions, and activation functions.

Layer Name	Total Number of Trainable Parameters
Conv2-D Concatenate MaxPooling 2-D Conv2-D Transpose	84,632,001

**Table 5 biomimetics-10-00509-t005:** The details of optimal parameters for training the CS-Ensemble Net.

Parameters	Search Scope	Optimal Value
Optimizer of first part	Adam	Adam
Cost function of first part	MAE, Dice Loss	Dice Loss
CS ratio	5%, 25%, 80%, 100%	80%
Learning rate of first part of Ensemble Net	0.1, 0.01, 0.001	0.001
Shortcut path of second part	Simple, CNN	CNN
Optimizer of MA	Adam	Adam
Learning rate of second part of Ensemble Net	0.01, 0.001, 0.0001, 0.00001	0.0001
Number of transposed 2D-convolution layers of decoders	1, 2, 3, 4	3
Number of 2D-convolution layers of encoders	1, 2, 3, 4	3

**Table 6 biomimetics-10-00509-t006:** Segmentation results using CBV, CBF, MTT, and Tmax in CS ratio of 80%.

CTP Mode	Methods	Accuracy (%)	Sensitivity (%)	Dice-Coeff (%)	Mean-IoU (%)
**CBV**	MultiresUNet	73.12	70.09	71.12	65.59
	Ensemble Net	84.03	81.95	82.23	77.84
	CS-Ensemble Net	89.09	86.64	87.65	84.56
**CBF**	MultiresUNet	70.14	68.52	68.96	63.79
	Ensemble Net	79.96	76.89	78.56	73.84
	CS-Ensemble Net	85.62	84.23	84.51	82.09
**MTT**	MultiresUNet	71.45	69.76	70.64	63.79
	Ensemble Net	80.62	78.89	78.96	77.59
	CS-Ensemble Net	87.93	85.29	86.75	83.09
**Tmax**	MultiresUNet	73.43	72.44	72.69	70.86
	Ensemble Net	82.34	80.05	81.96	79.92
	CS-Ensemble Net	89.34	86.91	87.73	85.64
**[CBV, CBF, MTT, Tmax, CTSlice]**	MultiresUNet	76.52	73.09	75.12	71.19
	Ensemble Net	86.61	84.13	84.98	77.67
	CS-Ensemble Net	92.43	90.14	91.66	86.16

**Table 7 biomimetics-10-00509-t007:** Segmentation results under varying CS-ratios, showing performance metrics across the full test set using 10-fold cross-validation.

CS-Ratio	CTP Mode	Accuracy (%)	Sensitivity (%)	Dice-Coeff (%)	Mean-IoU (%)
5%	[CBV, CBF, MTT, Tmax, CTSlice]	86.69	85.09	85.41	81.99
20%		90.03	87.77	88.02	86.35
80%		92.43	91.30	91.83	87.82

**Table 8 biomimetics-10-00509-t008:** The segmentation results of noisy CT slices.

CTP Mode	Methods	Accuracy (%)	Sensitivity (%)	Dice-Coeff (%)	Mean-IoU (%)
CBV with noise	MultiresUnet	76.52	74.18	75.93	73.03
	Ensemble Net	80.15	77.29	79.02	76.04
	CS-Ensemble Net	85.02	83.98	83.99	84.2

**Table 9 biomimetics-10-00509-t009:** Comparison with recent models.

Methods	Accuracy (%)	Sensitivity (%)	Dice-Coeff (%)	Mean-IoU (%)
CS-Ensemble Net	92.43	91.30	91.83	87.82
Ensemble Net	80.15	77.29	79.02	76.04
MultiresUnet [18]	76.52	74.18	75.93	73.03
SLNet [14]	-	64	54	-
R2U-Net [27]	-	60	52	-
FCN [28]	-	57	49	-
Multi-scale U-Net [29]	-	55	55	-

## Data Availability

The dataset used in this study is publicly available in this address: https://www.isles-challenge.org (accessed on 29 July 2025).

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
