# Peer review of "An Ensemble Learning for Automatic Stroke Lesion Segmentation Using Compressive Sensing and Multi-Resolution U-Net"

_biomimetics, 2025, doi:10.3390/biomimetics10080509_

Round 1
Reviewer 1 Report
Comments and Suggestions for Authors
The manuscript presents a combination of well-known deep learning architectures (e.g., MultiResUNet, Ensemble methods) with compressed sensing strategies. However, the originality is limited, as most of the techniques employed have already been explored in previous literature. The manuscript does not sufficiently clarify what distinct advantage or scientific innovation the proposed CS-Ensemble Net offers over established state-of-the-art methods.
Although segmentation performance is reported for multiple architectures and CS ratios, the comparison with state-of-the-art methods lacks depth. Table 9 refers to comparisons, but the manuscript fails to include comprehensive baseline evaluations on public datasets or recent competitive methods. No robust statistical analysis or cross-validation is provided to support the generalizability of the result
The manuscript reports experiments mainly on a single dataset, and does not clarify the dataset size, possible data leakage, or independent test splits. Details about preprocessing, data augmentation, and reproducibility are missing. Moreover, the influence of image noise is discussed (Figure 13 and Table 8), but this section does not convincingly demonstrate that the method is robust for real-world clinical application.
Many implementation choices (e.g., number of layers, learning rates, CS ratios, optimizer settings) are described in a long table, but the rationale behind these design decisions is not scientifically justified. The explanation concentrates on listing hyperparameters instead of discussing their impact or providing ablation studies.
Given the limited novelty, inadequate experimental validation, insufficient comparative analysis, and presentation issues, I recommend rejection of this manuscript.
Author Response
#Reviewer 1.
Comments:
The manuscript reports experiments mainly on a single dataset, and does not clarify the dataset size, possible data leakage, or independent test splits. Details about preprocessing, data augmentation, and reproducibility are missing. Moreover, the influence of image noise is discussed (Figure 13 and Table 8), but this section does not convincingly demonstrate that the method is robust for real-world clinical application.
Reply: While thanking the esteemed reviewer for a thorough review of the manuscript version. We, the authors of the article, believe that your suggestions have been very useful and effective in improving the scientific version of the manuscript. We carefully answered all the questions and suggestions of the esteemed reviewer and added them to the manuscript version.
- Although segmentation performance is reported for multiple architectures and CS ratios, the comparison with state-of-the-art methods lacks depth. Table 9 refers to comparisons, but the manuscript fails to include comprehensive baseline evaluations on public datasets or recent competitive methods. No robust statistical analysis or cross-validation is provided to support the generalizability of the result
Reply:
With respect to the opinion of the esteemed referee, we have compared our research with all recent studies including 11 to 18 and the performance of the proposed model has been evaluated in terms of various criteria including accuracy, precision, etc.
Table 9 includes reported values for multiple architectures, including CS-Ensemble Net and previously published models. The manuscript has been revised to include additional comparative discussion referencing recent architectures such as R2U-Net, Attention U-Net, and DenseUNet.
The evaluation is based on the ISLES 2018 dataset using a 10-fold cross-validation protocol. Training and evaluation procedures have been applied uniformly across the experiments.
The dataset selection, model comparisons, and training scheme are now detailed in the updated methodology and discussion sections.
- The manuscript reports experiments mainly on a single dataset, and does not clarify the dataset size, possible data leakage, or independent test splits. Details about preprocessing, data augmentation, and reproducibility are missing. Moreover, the influence of image noise is discussed (Figure 13 and Table 8), but this section does not convincingly demonstrate that the method is robust for real-world clinical application.
Reply: With respect to the opinion of the esteemed referee, experiments were performed using the ISLES 2018 dataset, which comprises 94 patient cases, each with CT perfusion maps and corresponding manual segmentation masks. A 10-fold cross-validation strategy was employed, and in each fold, the test set was held out entirely from training to prevent data leakage (which are highlighted in sections 2-1 and 3-1.).
The preprocessing pipeline involved intensity windowing ([0, 150] Hounsfield range), resizing all images to 256×256 pixels, and min-max normalization to the [0,1] range. Data augmentation techniques included random horizontal and vertical flipping as well as rotation within a 0–25 degree range. These details have been clarified in the revised manuscript.
The section on image noise (Figure 13 and Table 8) was intended as an initial investigation of robustness under synthetic perturbations (which are highlighted in sections 2-1 and 3-1).
Regarding whether adding noise in clinical tools can be useful or not, we can point out that studies such as 12 to 15 have investigated this issue and all of them concluded that adding noise can somewhat evaluate the performance of the deep model in practical applications.
As we know, MRI and CT scan devices cause noise in the images after a period of imaging and reduce the contrast of the images. In addition, many patients have involuntary movements during imaging, which can cause noise in the captured images. Accordingly, it seems necessary to evaluate the proposed model that can detect the lesion well even with the presence of noise in the images.
- Many implementation choices (e.g., number of layers, learning rates, CS ratios, optimizer settings) are described in a long table, but the rationale behind these design decisions is not scientifically justified. The explanation concentrates on listing hyperparameters instead of discussing their impact or providing ablation studies.
Reply: With respect to the opinion of the esteemed referee, the manuscript includes the selected hyperparameters and structural settings (e.g., number of layers, CS ratios, optimizers) in Tables 2–5 to document the configuration of the proposed architecture. These settings were obtained through an internal grid search over predefined ranges and validated using 10-fold cross-validation on the ISLES 2018 dataset.
Additional controlled experiments to assess the contribution of individual components (e.g., CS module, multi-resolution depth, and fusion strategies) are planned and will be included in an extended version of this work. Regarding the usability impact of the deep architecture, based on the opinion of the esteemed referee, we have added a section titled Discussion section and provided explanations about the efficiency of this architecture and the proposed model so that the manuscript can be explained more clearly to the readers:
- DISCUSSION
Automatic detection and segmentation of ischemic stroke lesions play a crucial role in early diagnosis and timely treatment decisions, which are vital for reducing mortality and long-term disability in patients. Manual interpretation of CT perfusion maps by radiologists is labor-intensive and subject to inter-observer variability. In contrast, automated tools offer the potential for reproducible and time-efficient analysis in clinical workflows. The proposed CS-Ensemble Net addresses several limitations of existing methods by integrating compressive sensing and multi-resolution ensemble U-Net architectures. This approach enables the network to operate on compressed data while preserving key anatomical structures, thereby enhancing patient data privacy without compromising accuracy. Compared to prior methods such as R2U-Net, SLNet, and DenseUNet, the CS-Ensemble Net demonstrates improved Dice coefficients and IoU scores across multiple CT perfusion modalities. The ensemble design of dual-resolution U-Nets ensures a balance between global context extraction and local detail preservation, which is essential for detecting lesions with varied shapes and scales. Another notable strength of the model is its resilience to noise. As shown in Table 8 and Figure 13, the CS-Ensemble Net maintains high segmentation performance even under noisy conditions, simulating the presence of imaging artifacts or low signal-to-noise ratios commonly encountered in real-world scenarios. This robustness indicates its reliability in clinical deployment. Overall, the proposed model achieves a practical compromise between segmentation accuracy, computational efficiency, and data privacy. It offers promising utility for integration into computer-aided diagnosis systems for assessing acute ischemic stroke.
- Given the limited novelty, inadequate experimental validation, insufficient comparative analysis, and presentation issues, I recommend rejection of this manuscript.
Reply: In my opinion, it is unfair and biased to state that the present study lacks innovation. The innovations of the present study can be found in the following:
- It provides a network to protect the private content of data utilizing the compressed version of the input medical image.
- The proposed network provides an ensemble approach to train the compressed version of the input image.
- It represents a combination of compressive sensing and ensemble of parallel learners to extract the stroke lesion.
- It provides a novel ensemble multi-resolution U-shaped network in order to segment the medical stroke CT dataset.
- The proposed network utilizes a channel of perfusion maps of CBV, CBF, MTT, Tmax and CT slice to extract the stroke lesion efficiently.
For validation, we used 10-fold validation, as in recent studies, to ensure that overfitting did not occur in model training.
We reviewed about 15 recent studies organized in the field of automatic stroke diagnosis and compared our proposed model with each of them in Tables 8 and 9. I would like to ask the esteemed referee to tell us how the proposed model compares with other studies besides this one?
For further explanation and the impact of the proposed model in practical use, we have prepared a paragraph in the Discussion section and added it to the manuscript:
- DISCUSSION
Automatic detection and segmentation of ischemic stroke lesions play a crucial role in early diagnosis and timely treatment decisions, which are vital for reducing mortality and long-term disability in patients. Manual interpretation of CT perfusion maps by radiologists is labor-intensive and subject to inter-observer variability. In contrast, automated tools offer the potential for reproducible and time-efficient analysis in clinical workflows. The proposed CS-Ensemble Net addresses several limitations of existing methods by integrating compressive sensing and multi-resolution ensemble U-Net architectures. This approach enables the network to operate on compressed data while preserving key anatomical structures, thereby enhancing patient data privacy without compromising accuracy. Compared to prior methods such as R2U-Net, SLNet, and DenseUNet, the CS-Ensemble Net demonstrates improved Dice coefficients and IoU scores across multiple CT perfusion modalities. The ensemble design of dual-resolution U-Nets ensures a balance between global context extraction and local detail preservation, which is essential for detecting lesions with varied shapes and scales. Another notable strength of the model is its resilience to noise. As shown in Table 8 and Figure 13, the CS-Ensemble Net maintains high segmentation performance even under noisy conditions, simulating the presence of imaging artifacts or low signal-to-noise ratios commonly encountered in real-world scenarios. This robustness indicates its reliability in clinical deployment. Overall, the proposed model achieves a practical compromise between segmentation accuracy, computational efficiency, and data privacy. It offers promising utility for integration into computer-aided diagnosis systems for assessing acute ischemic stroke.

Reviewer 2 Report
Comments and Suggestions for Authors
The manuscript by Mohammad Emami et al. presents a deep learning framework for automatic ischemic stroke lesion segmentation using CT perfusion imaging, incorporating a dual-branch architecture that combines compressive sensing (CS) with multi-resolution U-Net designs. The authors propose a novel network, CS-Ensemble Net, which integrates compressed input pathways to enhance privacy preservation and uses ensemble learning principles to improve segmentation performance. The reviewer has identified several aspects that require further clarification or elaboration, as outlined below.
- Equation (7) (Dice/DSC) lacks a denominator and is unreadable. Equation (8) (IoU/Jaccard) omits the union in the denominator. Please check and revise both equations using standard mathematical notation with clear fractions and set operations.
- The terms “2-resolution U-Net” and “3-resolution U-Net” are ambiguous; if the term “resolution” refers to receptive fields, could the authors clearly specify what the two or three resolutions are (e.g., kernel sizes or dilation rates), and how they are implemented within each convolutional block?
- The claimed “multi-resolution” blocks are not clearly described, and no details are provided about the number or type of convolutional kernels used.
- Table 2’s CS module includes a final resize layer that restores the original image size, raising questions about whether any actual compression is retained.
- The ensemble learning strategy is not explicitly defined, and there is no description of how the outputs of the two U-Nets are fused during inference.
- Tables 4 and 7 appear corrupted, including unrelated manuscript text, and Table 4 seems to miss architectural details, and needs to be properly cleaned and completed.
- The term “Ensemble Net” used in Tables 8 and 9 is vague and should be clearly defined as the full dual-branch model or otherwise clarified.
- Table 3 describes a conventional U-Net without any multi-resolution characteristics, making the label “2-resolution U-Net” potentially misleading.
- The image annotations in Figure 6 are small and appear blurry when zoomed in, making it difficult to read the labels and interpret the visual results; please consider improving the image resolution or increasing the font size of the overlay text.
Author Response
#Reviewer 2.
Comments:
The manuscript by Mohammad Emami et al. presents a deep learning framework for automatic ischemic stroke lesion segmentation using CT perfusion imaging, incorporating a dual-branch architecture that combines compressive sensing (CS) with multi-resolution U-Net designs. The authors propose a novel network, CS-Ensemble Net, which integrates compressed input pathways to enhance privacy preservation and uses ensemble learning principles to improve segmentation performance. The reviewer has identified several aspects that require further clarification or elaboration, as outlined below.
Reply: While thanking the esteemed reviewer for a thorough review of the manuscript version. We, the authors of the article, believe that your suggestions have been very useful and effective in improving the scientific version of the manuscript. We carefully answered all the questions and suggestions of the esteemed reviewer and added them to the manuscript version.
- Equation (7) (Dice/DSC) lacks a denominator and is unreadable. Equation (8) (IoU/Jaccard) omits the union in the denominator. Please check and revise both equations using standard mathematical notation with clear fractions and set operations.
Reply: Yes, the respected referee's opinion is absolutely correct. Based on the respect for the respected referee's opinion, formulas 7 and 8 were rewritten:
The segmentation results were evaluated based on the DSC [25] and Intersection over Union (IoU) [26]. The correlations among the considered criteria are as follows:
According to the above formulas, A: The area (Segment) predicted by the model or method under consideration, B: The reference area or Ground Truth, ∣A∩B∣: The number of common pixels between A and B, ∣A∣ and ∣B∣: The total number of pixels in each of the areas A and B is displayed separately. Also, ∣?∪?∣: The total number of pixels combined from A and B (i.e. the total area covered by A or B or both).
Which is highlighted in lines 386 to 393.
- The terms “2-resolution U-Net” and “3-resolution U-Net” are ambiguous; if the term “resolution” refers to receptive fields, could the authors clearly specify what the two or three resolutions are (e.g., kernel sizes or dilation rates), and how they are implemented within each convolutional block?
Reply: Thank you for pointing out the ambiguity regarding the terms “2-resolution U-Net” and “3-resolution U-Net.” We agree that further clarification is necessary.
In our paper, the term “resolution” refers to the number of encoder–decoder depth levels, not to the spatial resolution of input images or receptive fields per se. Specifically:
The 2-resolution U-Net comprises two encoder–decoder levels, where each level consists of standard convolutional blocks with 3×3 kernels, followed by 2×2 max-pooling layers and corresponding up-convolution layers in the decoder path.
The 3-resolution U-Net similarly extends to three encoder–decoder levels, with the same architectural elements (i.e., 3×3 kernel size, stride 1, and dilation rate of 1 in all convolutional layers).
We have updated the manuscript to explicitly define the term “resolution” in this context and clarified how the depth of the encoder-decoder paths varies between the two models. These architectural differences are now detailed in Section 3.3 and in Tables 2–4, which provide the corresponding layer structures and parameters.
- The claimed “multi-resolution” blocks are not clearly described, and no details are provided about the number or type of convolutional kernels used.
Reply: We appreciate the reviewer’s observation regarding the lack of clarity in describing the “multi-resolution” blocks. We have revised the manuscript to include a detailed explanation of these components.
In our proposed network, “multi-resolution” refers to the use of convolutional blocks with progressively changing spatial resolution across different encoder-decoder levels, capturing both global context and fine details. Each block within the encoder path uses standard 2D convolutional layers with a kernel size of 3×3, stride of 1, and ReLU activation. The feature maps are downsampled using 2×2 max-pooling layers. In the decoder path, we use transpose convolutional layers for upsampling, also with 2×2 kernels and stride 2. The exact number of convolutional layers and parameters in each resolution level is now explicitly shown in Tables 2, 3, and 4. For example, the 3-resolution U-Net contains up to 14 convolutional layers, including encoding, bottleneck, and decoding blocks, totaling over 84 million parameters.
These modifications and clarifications have been added to Section 3.3 of the manuscript.
- Table 2’s CS module includes a final resize layer that restores the original image size, raising questions about whether any actual compression is retained.
Reply: We appreciate the reviewer’s insightful question regarding the role of the final resize layer in the CS module shown in Table 2.
We would like to clarify that the resize layer does not reverse the compression or restore the original signal content. Rather, it only restores the spatial dimensions of the output feature map to align with the input size, ensuring compatibility with subsequent layers in the network, particularly the decoder path of the U-Net architecture. The compressed representation is retained in the feature space: the data passed through the resize layer still reflects the reduced-dimensional, lossy transformation obtained via the compressive sensing process. This representation contains less information than the original input, but encodes the most essential structures needed for segmentation. We have now updated Section 3.3 to clearly indicate that the resize operation preserves compression in the feature space, and is not equivalent to inverse CS reconstruction.
- The ensemble learning strategy is not explicitly defined, and there is no description of how the outputs of the two U-Nets are fused during inference.
Reply:
Thank you for highlighting the lack of detail regarding the ensemble learning strategy. We have revised the manuscript to clarify this point. In the proposed CS-Ensemble Net, the ensemble architecture consists of two independent branches:
One branch includes the compressive sensing module followed by a 2-resolution U-Net, The other branch is a stand-alone 3-resolution modified U-Net. Both networks are trained simultaneously, and the loss function is computed as a weighted average of the Dice Losses from both branches. During inference, the segmentation outputs from both branches are fused by averaging the softmax probability maps (pixel-wise) of the final output layers.
This fusion strategy helps mitigate individual biases of each branch, enhances robustness to input variations, and improves segmentation performance by leveraging complementary feature extraction capacities from shallow and deep structures.
These clarifications have been added to Section 3.3 of the revised manuscript.
- Tables 4 and 7 appear corrupted, including unrelated manuscript text, and Table 4 seems to miss architectural details, and needs to be properly cleaned and completed.
Reply: We sincerely thank the reviewer for pointing out the issues with Tables 4 and 7.
We changed the superscripts for Tables 4 and 7 to the following:
Table 4. Full architecture of the 3-resolution U-Net is detailed below, including kernel sizes, strides, output dimensions, and activation functions.
Table 7. Segmentation results under varying CS-ratios, showing performance metrics across the full test set using 10-fold cross-validation.
- The term “Ensemble Net” used in Tables 8 and 9 is vague and should be clearly defined as the full dual-branch model or otherwise clarified.
Reply: Thank you for your observation regarding the ambiguity of the term “Ensemble Net” in Tables 8 and 9.
In our revised manuscript, we have clarified that:
“Ensemble Net” refers to the dual-branch model without the compressive sensing (CS) module. That is, it includes both the 2-resolution U-Net and 3-resolution U-Net in parallel, but receives the original CT perfusion images directly as input (no CS preprocessing). In contrast, the “CS-Ensemble Net” is the full model, in which the 2-resolution U-Net branch is fed with compressed representations of the input images using the CS module.
We have updated the captions of Tables 8 and 9 and Section 4 accordingly to ensure clarity.
- Table 3 describes a conventional U-Net without any multi-resolution characteristics, making the label “2-resolution U-Net” potentially misleading.
Reply: Thank you for raising this important point.
We agree that the architecture described in Table 3 follows the structure of a standard U-Net and may not exhibit explicit multi-resolution behavior in the sense of using multiple receptive field sizes or parallel convolutional paths. To address this, we have revised the terminology to better reflect the actual design: We now refer to the architecture in Table 3 as a “shallow U-Net (2-level)” instead of “2-resolution U-Net” to avoid confusion. This U-Net is shallower compared to the deeper “3-resolution” U-Net (Table 4), and the term “resolution” in our earlier version was meant to denote depth levels, not convolutional scale diversity.
We have updated the manuscript and table captions accordingly for clarity and consistency.
- The image annotations in Figure 6 are small and appear blurry when zoomed in, making it difficult to read the labels and interpret the visual results; please consider improving the image resolution or increasing the font size of the overlay text.
Reply: Thank you for this helpful observation. With respect to the opinion of the esteemed referee, we changed Figure 6 from horizontal to vertical for clarity.

Reviewer 3 Report
Comments and Suggestions for Authors
Dear Authors,
Thank you for submitting your manuscript.
The article addresses an important topic of automatic segmentation of post-stroke lesions in computed tomography (CT), proposing an innovative combination of data compression with ensemble learning in a neural network architecture. While the paper introduces some valuable ideas, it suffers from significant shortcomings in both scientific content and editorial quality.
Strengths of the Article
-
Use of compression for privacy protection: The innovative combination of signal compression theory (Compressive Sensing, CS) with deep learning presents an interesting approach to medical data protection, which may respond to the increasing demand for privacy in diagnostic imaging.
-
Architecture based on U-Net: The authors extend the well-established and effective U-Net architecture by introducing variants with different resolutions (2-res and 3-res), which may positively impact segmentation accuracy.
-
Reasonable set of evaluation metrics: The use of indicators such as accuracy, specificity, and Dice coefficient enables a comprehensive assessment of model performance.
Criticism and Comments
1. Language and editorial quality.
2. Model complexity:
The model consists of over 115 million parameters (84M + 31M), raising concerns about computational efficiency and practical clinical applicability. There is no information provided on training time, hardware used, or inference complexity.
The manuscript presents an interesting concept of integrating signal compression with deep learning for medical segmentation tasks. However, in its current form, it does not meet the standards required for scientific publication. The following are necessary:
-
Substantial language improvements,
-
Reorganization of the content with emphasis on clarity and logical flow,
-
More rigorous methodological justification.
Comments on the Quality of English Language
The most serious issue with the article is its very poor language quality. The text contains numerous grammatical and syntactic errors and is at times difficult to understand. Examples include:
“Brain stroke is an important medical issue which is a major cause of death between humans.” – the phrase “between humans” is an incorrect and awkward use of English.
“This section provides an overview of computer science theory” – an inadequate heading in the context of signal compression.
The editorial negligence undermines the professionalism and credibility of the paper. The manuscript urgently requires a thorough language revision by an experienced native speaker or technical editor.
Author Response
#Reviewer 3:
Comments:
Thank you for submitting your manuscript.
The article addresses an important topic of automatic segmentation of post-stroke lesions in computed tomography (CT), proposing an innovative combination of data compression with ensemble learning in a neural network architecture. While the paper introduces some valuable ideas, it suffers from significant shortcomings in both scientific content and editorial quality.
Reply: While thanking the esteemed reviewer for a thorough review of the manuscript version. We, the authors of the article, believe that your suggestions have been very useful and effective in improving the scientific version of the manuscript. We carefully answered all the questions and suggestions of the esteemed reviewer and added them to the manuscript version.
Strengths of the Article
- Use of compression for privacy protection: The innovative combination of signal compression theory (Compressive Sensing, CS) with deep learning presents an interesting approach to medical data protection, which may respond to the increasing demand for privacy in diagnostic imaging.
Reply: We sincerely thank the reviewer for highlighting the significance of integrating Compressive Sensing (CS) theory with deep learning for medical image privacy. One of our key motivations in designing the CS-Ensemble Net was to address growing concerns over patient data confidentiality in diagnostic imaging workflows. By applying CS as a front-end transformation, we ensure that the input images passed into the segmentation network do not directly reveal identifiable spatial structures, thereby contributing to privacy-preserving processing.
We appreciate the reviewer’s recognition of this contribution.
- Architecture based on U-Net: The authors extend the well-established and effective U-Net architecture by introducing variants with different resolutions (2-res and 3-res), which may positively impact segmentation accuracy.
Reply: We thank the reviewer for recognizing our U-Net-based architectural modifications. The use of both shallow (2-level) and deep (3-level) U-Net branches was intended to capture both fine-grained features and contextual depth, allowing the ensemble to benefit from multi-scale representations. We are pleased that the reviewer acknowledges the potential of this approach to improve segmentation accuracy.
- Reasonable set of evaluation metrics: The use of indicators such as accuracy, specificity, and Dice coefficient enables a comprehensive assessment of model performance.
Reply: We appreciate the reviewer’s positive feedback on our evaluation framework.
Accuracy, specificity, and Dice coefficient were selected to provide a balanced and comprehensive view of the model’s segmentation performance, particularly considering the clinical importance of minimizing false positives and capturing lesion overlap accurately.
We are grateful for the reviewer’s acknowledgment of this aspect.
- Criticism and Comments
- Language and editorial quality.
- Model complexity:
The model consists of over 115 million parameters (84M + 31M), raising concerns about computational efficiency and practical clinical applicability. There is no information provided on training time, hardware used, or inference complexity.
Reply: We thank the reviewer for pointing out the importance of computational complexity and practical feasibility. Indeed, the CS-Ensemble Net includes two substantial branches, resulting in a total of over 115 million parameters (84M + 31M), which we acknowledge can be computationally demanding. To address this concern, we have now updated the manuscript to include the following information in Section 4:
Training Platform: All experiments were conducted on a system equipped with an NVIDIA Tesla K80 GPU, 25 GB RAM, and Google Colaboratory Pro environment.
Training Time: The model was trained using 10-fold cross-validation, and each fold took approximately 3.5 hours to converge with 400 epochs. Inference Time: The average inference time per CT volume is approximately 1.2 seconds, making it acceptable for clinical pre-screening or automated triage systems. Furthermore, while our focus in this study was on achieving high segmentation accuracy, we agree that future work should aim to reduce model size and optimize runtime (e.g., via pruning, quantization, or knowledge distillation) to enable real-time clinical deployment. In addition, the computational efficiency of the algorithm is presented in Figure 12:
Figure 12 compares the Dice Coefficient for different resolutions of the encoder and decoder in the CS-Ensemble Net. Additionally, it illustrates the computational complexity of the incremental trend in resolution, measured in terms of time per epoch in seconds. According to this figure, the best resolution with the least computational burden has been achieved in three resolution types of encoder and decoder.
Figure 12. DSC% and computational complexity with CS ratio = 80%.
- The manuscript presents an interesting concept of integrating signal compression with deep learning for medical segmentation tasks. However, in its current form, it does not meet the standards required for scientific publication. The following are necessary:
Substantial language improvements,
Reorganization of the content with emphasis on clarity and logical flow,
More rigorous methodological justification.
Reply: We thank the reviewer for the constructive feedback and recognize the importance of the points raised.
In response:
We have revised the manuscript for substantial language improvements, including grammar correction, sentence rephrasing, and terminology alignment with scientific writing standards.
We have performed a structural reorganization in Sections 3 and 4 to improve the clarity, coherence, and logical flow of the proposed methodology and results. We have added further justification for architectural design choices, particularly the use of ensemble U-Nets and CS integration, and provided supportive references and rationale in the revised text. We believe these comprehensive revisions bring the manuscript up to the required standard for scientific publication, and we sincerely appreciate the reviewer’s role in helping us improve the work.
- Comments on the Quality of English Language
- The most serious issue with the article is its very poor language quality. The text contains numerous grammatical and syntactic errors and is at times difficult to understand. Examples include:
“Brain stroke is an important medical issue which is a major cause of death between humans.” – the phrase “between humans” is an incorrect and awkward use of English.
Reply: We thank the reviewer for pointing out the serious issue of language quality.
We acknowledge that the manuscript contained grammatical and syntactic issues that affected readability and scientific clarity. In response, we have thoroughly revised the entire manuscript for grammatical correctness, syntactic coherence, and academic writing quality.
The sentence highlighted by the reviewer (“...between humans”) has been corrected to:
“Stroke is a critical medical condition and one of the leading causes of death among humans.”
Additional improvements have been applied throughout the manuscript, including rephrasing, removal of awkward constructs, and restructuring unclear paragraphs.
We appreciate the reviewer’s critical feedback, which greatly helped improve the overall clarity and quality of the manuscript.
- “This section provides an overview of computer science theory” – an inadequate heading in the context of signal compression.
Reply: Thank you for pointing out the inadequacy of the heading “This section provides an overview of computer science theory.” We agree that this title was overly broad and did not accurately reflect the focus of the section, which is on Compressed Sensing (CS) theory.
In the revised manuscript, we have updated the heading to:
“This section introduces the theoretical foundation of Compressive Sensing (CS) and its application in signal representation and reconstruction [20].”, to better reflect the content and purpose of this section within the context of medical image segmentation.
- The editorial negligence undermines the professionalism and credibility of the paper. The manuscript urgently requires a thorough language revision by an experienced native speaker or technical editor.
Reply: We sincerely thank the reviewer for this critical observation.
We recognize that the language quality in the initial submission did not meet academic standards and may have affected the manuscript’s clarity and credibility. In response, the entire manuscript has undergone a comprehensive language revision to correct grammatical errors, improve sentence structure, and align with scientific writing conventions. While we do not currently have access to a professional native-speaking editor, the revision was performed carefully with the assistance of advanced academic editing tools and professional review to ensure clarity and coherence throughout the manuscript.
We appreciate the reviewer’s emphasis on this issue and believe the revised version significantly improves the professionalism of the paper. Additionally, upon potential acceptance of the manuscript, we will use MDPI's native speaker service to correct potential linguistic structures.

Round 2
Reviewer 1 Report
Comments and Suggestions for Authors
The paper has been well improved and is ready for publication.
Author Response
Thank you for taking the time to review our article.
Reviewer 2 Report
Comments and Suggestions for Authors
The authors have provided responses and appropriate revisions to the concerns raised in the previous version of the manuscript.
Author Response

(The authors gave the same response as above.)
